# PCA-MRM Model to Forecast TEC at Middle Latitudes

**Anna L. Morozova *** , **Teresa Barata** and **Tatiana Barlyaeva**

Instituto de Astrofísica e Ciências do Espaço, Observatório Geofísico e Astronómico da Universidade de Coimbra, University of Coimbra, 3040-004 Coimbra, Portugal; mtbarata@gmail.com (T.B.); tvbarlyaeva@gmail.com (T.B.)
* Correspondence: annamorozovauc@gmail.com

**Abstract:** The total electron content (TEC) over the Iberian Peninsula was modelled using PCA-MRM models based on decomposition of the observed TEC series using the principal component analysis (PCA) and reconstruction of the daily modes' amplitudes by a multiple linear regression model (MRM) using space weather parameters as regressors. The following space weather parameters are used: proxies for the solar UV and XR fluxes, number of the solar flares of different classes, parameters of the solar wind and of the interplanetary magnetic field, and geomagnetic indices. Time lags of 1 and 2 days between the TEC and space weather parameters are used. The performance of the PCA-MRM model is tested using data for 2015, both geomagnetically quiet and disturbed periods. The model performs well for quiet days and days with solar flares but without geomagnetic disturbances. The MAE and RMSE metrics are of the order of 3–5 TECu for daytime and ~2 TECu for night-time. During geomagnetically disturbed periods, the performance of the model deteriorates but only for daytime: MAE and RMSE are of the order of 4–6 TECu and can rise to ~13 TECu for the strongest geomagnetic storms.

**Keywords:** TEC model; middle latitudes; space weather; principal component analysis

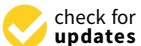



## 1. Introduction

Space Weather (a group of phenomena observed in the near-Earth space and atmosphere that are related to variations of the solar activity) is one of the main drivers of ionospheric disturbances which, in turn, can drastically affect the quality of the signal between a ground-based device and the GNSS (global navigation satellite system) satellites. Forecasting and early warning for potentially dangerous space weather events are essential for the improvement of the quality of the GNSS-based services [1]. On the other hand, the GNSS receivers themselves are a reliable source of ionospheric data. Data from GNSS receivers can be utilised to estimate such widely used ionospheric parameters as the total electron content (TEC), scintillation indices (S4, $\sigma_{\varphi}$) and ROTI (rate of TEC index).

The forecasts of the ionospheric parameters are based on the understanding of the ionosphere reaction to different forcings. Empirical (based on the observational data and their statistical analysis) models for ionospheric response to external forcings (e.g., solar flares and geomagnetic storms) are being developed by different research groups. The principal component analysis (PCA, also known as the empirical orthogonal functions (EOF) or the natural orthogonal components) and different kinds of regression analyses are often used to model and forecast TEC variations using some space weather parameters as regressors or predictors. Below we give a short review on the forecasting quality of such models, though we do not claim that this review is complete.

The authors of [2,3] proposed EOF-based models to forecast monthly median TEC in the form of regional ionosphere maps (RIMs) for China using the F10.7 solar index as predictor. Their models provided TEC forecasts with the standard deviation (SD) of 1–4 TECu (1 TECu = $10^{16}$ el/m$^2$), with lower SD during solar activity minima and higher SD for years of solar activity maxima.

In [4], neural network (NN)-based models were proposed that used F10.7 index, sunspot numbers (SSN) and a set of proxies for the solar UV flux (e.g., Mg II index) as TEC predictors. Their best model provided the root mean squared error RMSE = 3 TECu. They also showed a delayed response of TEC variations to the solar forcing: best results were obtained if SSN, F10.7, and solar UV proxies lagged backward relative to TEC series by 1–2 days.

The authors of [5] used an EOF-based model to reconstruct global ionosphere maps (GIMs). They decomposed TEC observations into EOF functions that change with local time and dip latitude to represent the diurnal variation and spatial distribution of the original data and fitted their amplitude coefficients (that indicate the long-term temporal fluctuations) by F10.7 and geomagnetic indices Ap, AE and Dst. The model provided RMSE = 3–5% which for middle latitudes gives RMSE $\approx$ 2–3 TECu. In [6] PCA was also used to decompose GIM TEC variations (2007–2016) into several spatio-temporal modes fitting their amplitude coefficients by Ap and F10.7 indices. The ME (mean error) was in the range from −15 to 10 TECu and RMSE was in the range from below 5 to ~15 TECu for different years with the worst errors observed during 2015, the year of strongest geomagnetic storms of the solar cycle 24.

The daily mean TEC (GIMs) was modelled in [7] using SSN and F10.7 as predictors. Their climatological model showed that the maximal differences between the observed and modelled TEC values are associated with geomagnetic storms (~3.2 ± 1.5 TECu). The authors of [8] proposed a GIM model to study the effect of geomagnetic activity on the ionospheric conditions. They modelled the relative TEC (relative deviation from a 15-day median) using the Kp geomagnetic index as predictor (RMSE = 4.6 TECu).

Another EOF-based model by [9] was proposed to model TEC variations for geomagnetically quiet months with Ap and F10.7 indices as predictors of TEC variations: the mean average error MAE = 1.2–2.6 TECu. Storm-time TEC variations were modelled by [10] for mid-low latitudes using EOF and NN with F10.7 and A-type indices as predictors for TEC variations. The RMSE values were between 2 and 10 TECu for different storms with Dst $\leq$ −50 nT occurred between 2000 and 2015.

TEC variations at the European middle latitudes during quiet periods of 2015 were modelled by [11] using a number of models of different types. Their models that use space weather parameters as predictors give for the 40ºN latitude band RMSE = 3.08–3.82 TECu, ME between -0.4 and 0.1 TECu, the median error between −0.1 and 0.3 TECu and the maximum error MaxE = 12–25 TECu. Another model that was tested on the data of 2015 was made by [12] for Balkan region. They used Kp and F10.7 indices as representatives of the solar and geomagnetic activity. They also showed delayed response of the ionosphere to geomagnetic storms (lag of 1–2 days for Kp). RMSE were in the range from 2.45 to 3.13 TECu and different between the night- (RMSE = 1.34–1.84 TECu) and daytime (RMSE = 4.5–5.5 TECu).

The single-station TEC (measured at a single location) was modelled by [13] using the F10.7 index as a proxy for the solar/geomagnetic activity (regression models) with RMSE = 3.22–4.46 TECu for different locations.

Almost all TEC models mentioned above, except, to some extent, those presented in [5,6], use additionally the information on the day of a year (DOY) and the hour to be able to model TEC daily and seasonal variations. The climatological models and the model built for geomagnetically quiet periods show, in general, better performance (lower RMSE and other errors) than models that use all or geomagnetically disturbed periods. There are no significant differences in the performance of the models developed using data from a single station and those using RIMs/GIMs. Most of the models described above are built using several years of TEC and space weather data. Some models incorporate lags of 1–2 days between the TEC response to space weather variations. Most often used space weather parameters are F10.7 and Mg II indices as proxies for the solar UV irradiance and A-type indices to account for the geomagnetic activity variations. Overall, RMSE are in the

range from 2 to 15 TECu with higher RMSE values obtained for models that simulate TEC variations during geomagnetic storms.

Here we present the analysis of the performance of a new TEC model which is based both on the PCA and regression analyses. We use a different approach to model TEC daily and seasonal variations and a short (~30 days) time interval to develop a model. Additionally, we use a large set of space weather parameters as predictors of TEC variations. Our model does not distinguish between geomagnetically active and quiet days.

The paper is organised as follows: Section 1 gives a short summary of the performance of models developed to simulate TEC variations using space weather parameters as predictors; Section 2 describes the data used to build and test our TEC model; Section 3 describe methods utilised in our model and metric we used to test its performance; Sections 4 and 5 describe the proposed model and its performance, respectively; Section 6 presents discussion and main conclusions.

## 2. Data

To build and test our model we used following datasets from different ground and space missions.

### 2.1. Total Electron Content (TEC)

The TEC series is obtained at Lisbon (Portugal) using a GNSS receiver with the SCINDA software [14–17] that has been active from the November 2014 to the July 2019 in the Lisbon airport (38.8° N, 9.1° W) in the frame of the ESA Small ARTES Apps project SWAIR (Space Weather and GNSS monitoring services for Air Navigation). The installed equipment was a NovAtel EURO4 receiver with a JAVAD Choke-Ring antenna. The SCINDA software allows the conversion of the GNSS receiver data to TEC automatically.

The data originally of 1 min time resolution were averaged to obtain the 1 h series. In this work we used only data from January 1 to December 31 of 2015. The raw TEC data were processed using a "SCINDA-Iono" toolbox for MATLAB (see [18]) developed by our group and are available at [19]. This dataset is described in [18], and TEC variations related to the solar flares and geomagnetic disturbances of 2015 were analysed in [20].

The calibration procedure was not performed during the installation of this receiver. Therefore, we performed a calibration of the TEC records using the TEC data from the Royal Observatory of Belgium (ROB) as a reference. The Royal Observatory of Belgium (ROB) database (ftp:/gnss.oma.be/gnss/products/IONEX/, accessed on 14 February 2021) provides the vertical TEC maps in the IONEX format on a grid of 0.5° × 0.5° with 15 min time resolution [21]. The vertical TEC is estimated in near real-time from the GPS data of the EUREF Permanent Network (EPN). The similarity between our TEC series and series of ROB TEC were studied in [20]. To perform the calibration of our TEC series we used the ROB TEC data for the grid point most close to Lisbon (39° N, 9° W) using 1 h mean ROB TEC series. The calibration consisted of linear regression of our TEC series on the ROB TEC series with estimation of the scaling and shifting coefficients. It was performed individually for each of the 12 months. The scaling coefficients were in the range from 1.74 to 2.34 depending on the month allowing the conversion of the relative TEC SCINDA data [19] to the TEC data in TECu; shifting was not necessary. Since for Lisbon UT = LT, no time conversion was applied.

### 2.2. Space Weather Data

Preliminary tests of the PCA-MRM models as well as the results of [20,22] were used to define the set of space weather parameters (SWp) that are used as predictors for TEC variations. In particular, we prefer to use the Mg II index over the more often used F10.7 index [22]; also, the test models that used local $K_{COI}$ showed slightly better performance compared to ones that used the global Kp index. The separation of the number of flares of different classes (C, M and N) was proposed to find out what is more important for a TEC model: the total number of flares, the number of the most abundant flares (C-class or less)

or the number of moderate flares (M class); since in 2015 there was only one short-living X-class flare, the influence of the X flares on the model's performance was not studied.

Three types of SWp were used to forecast TEC variations:

1.  Parameters characterising the solar UV and XR fluxes:

    a.  a proxy for the solar UV irradiance: the Mg II composite series [23] based on the measurements of the emission core of the Mg II doublet (280 nm), hereafter Mg II;

    b.  a proxy for the solar XR irradiance: the measurements of the Solar EUV Experiment (SEE) for the NASA TIMED mission at the wavelength 0.5 nm, hereafter XR;

    c.  the daily number of the solar flares of the classes up to C, hereafter Number of C flares or C;

    d.  the daily number of the solar flares of the class M, hereafter Number of M flares or M;

    e.  total daily number of the solar flares of any class, hereafter Number of flares or N;

2.  Parameters characterising the interplanetary medium:

    a.  scalar of the interplanetary magnetic field (IMF), B in nT;

    b.  the X, Y and Z components of IMF in GSM frame, $B_X$, $B_Y$ and $B_Z$, respectively, all in nT;

    c.  the solar wind flow pressure (p in nPa), proton density (n in $n/cm^3$) and plasma speed (v in km/s).

3.  Geomagnetic indices:

    a.  the disturbance-time index Dst;

    b.  the global ap index;

    c.  the daily sums of the local K-index calculated from the horizontal component of the geomagnetic field measured at the Coimbra Magnetic Observatory, COI, Coimbra, Portugal, 40.2° N, 8.4° W, 99 m asl, $K_{COI}$;

    d.  the auroral electrojet index *AE* characterising the auroral activity in the polar regions.

The data on the solar wind properties were obtained from the OMNI database. The information about the solar flares observed during the analysed time interval was obtained through the NOAA National Geophysical Data Center (NGDC). Only flares that occurred during the local daytime were considered.

All SWp series used in the PCA-MRM model have 1 d time resolution and corresponding plots can be found in the Supplementary Material (SM), Figures S1–S5.

## 3. Methods

The PCA-MRM model is based on a combination of the principal component analysis (PCA) applied to the 1 h TEC series and multiple regression models (MRM) to forecast amplitudes of TEC daily variations using SWp as regressors. The quality of the forecast of the TEC series was characterised by a set of standard metrics.

### 3.1. PCA

The analysis of the TEC series was performed using the principal component analysis (PCA). The input dataset is used to construct a covariance matrix and calculate corresponding eigenvalues and eigenvectors. The eigenvectors are used to calculate principal components (PC) and empirical orthogonal functions (EOF). The eigenvalues allow us to estimate the explained variances of the extracted modes. PCs are orthogonal and conventionally non-dimensional, EOFs are in TECu. The full descriptions of the PCA method can be found in (e.g.,) [24–26].

The PCA input matrices were constructed in a way that each column contains 24 observations (means for every 1 h) for a specific day. Daily mean TEC values were removed before the series were submitted to PCA. The number of the columns, *L*, is equal to the

length of the studied interval in days. Thus, PCA allows us to obtain daily variations of different types as PCs and the amplitudes of those daily variations for each day of the studied time interval as corresponding EOFs. Consequently, PC series have 1 h time resolution (24 values each) and EOF series have 1 d time resolution (*L* values each). Only the 1st and 2nd PCA modes, PC1 and EOF1, and PC2 and EOF2, respectively, were used to reconstruct TEC variations [20].

### 3.2. Regression Models

The linear multiple regression models (MRM), i.e., linear regression models with not one but a number of regressors [27], were constructed to fit TEC series. In these MRMs, TEC-related parameters, i.e., the daily mean TEC (1 d mean TEC), and the EOF1 and EOF2 series, are dependent variables, and SWp are regressors.

MRMs were constructed to fit the data with all possible combinations of regressors, and the most appropriate model is selected as the one with minimal squared coefficient of multiple determination (see Section 3.3 and, e.g., [27]) adjusted for the number of degrees of freedom ($R_{adj}^2$). Thus, the selected model is built using a "best subset" of regressors ensuring that only those regressors that are most influential for a particular TEC series were selected.

According to our previous studies [20] and to some previous works e.g. [4,12] we used lags of 1 and 2 days between the TEC and SWp series (SWp lead). This setup allows us to use MRMs to forecast TEC as is explained in Section 4.1.

### 3.3. Metrics for the Forecast Quality

Similarities between the forecasted and observed TEC or other series were analysed using the Pearson correlation coefficients, *r*. The significance of the correlation coefficients was estimated using the Monte Carlo approach with artificial series constructed by the "phase randomization procedure" [28]. The obtained statistical significance (*p* value) considers the probability of a random series to have the same or higher absolute value of *r* as in the case of a tested pair of the original series.

The quality of the forecast was also estimated using the following parameters: the root mean squared error RMSE (Equation (1)), the explained variance ExpV (Equation (2)), the coefficient of determination $R^2$ (Equation (3)), the mean absolute error *MAE* (Equation (4)), the mean error ME (Equation (5)), the maximum error MaxE (Equation (6)), the *range* (defined as min/max values of a series), mean and median values of the differences between the forecasted and observed TEC values (ΔTEC and |ΔTEC| for absolute values of ΔTEC), the percentage of the forecasted series with ΔTEC in certain limits. The forecasting quality of a model is considered to be better if it had lower values of MAE and RMSE, higher values of r, $R^2$ and ExpV, and higher percentage of days with ΔTEC in a certain range.

$$RMSE = \sqrt{\Sigma(y_i - \hat{y}_i)^2 / N}, \tag{1}$$

$$ExpV = 1 - \sigma_{\Delta y}^2 / \sigma_y^2, \tag{2}$$

$$R^2 = 1 - \Sigma(y_i - \hat{y}_i)^2 / \Sigma(y_i - \overline{y})^2, \tag{3}$$

$$MAE = mean |\Delta TEC| = \Sigma|y_i - \hat{y}_i| / N, \tag{4}$$

$$ME = mean\ \Delta TEC = \Sigma(y_i - \hat{y}_i) / N, \tag{5}$$

$$MaxE = max(|y_i - \hat{y}_i|), \tag{6}$$

where $y_i$ and $\hat{y}_i$ are the observed and modelled series, respectively; $\overline{y}$ and $\sigma_y$ are the mean and SD for $y_i$, respectively; $\sigma_{\Delta y}$ is SD for the ΔTEC = $(y_i - \hat{y}_i)$ series; and N is the length of the series.

## 4. PCA-MRM Model for TEC Forecasting

### 4.1. General Description of the PCA-MRM TEC Model

The TEC forecasting using a PCA-MRM model consists of three stages (Figure 1). At the first stage the TEC data obtained for a certain time interval *L* (e.g., previous 20–40 days) are decomposed with PCA into several modes, each one representing a specific type of daily variation (PCs). Only two first modes, which have highest variance fractions and are responsible for most of the TEC variability, are used for further analysis [20]. One of the advantages of the model is that there is no need for any assumption on the phase and amplitude or seasonal/regional features of TEC daily variations: the daily variations of correct shapes are extracted automatically by PCA from the input TEC data. The examples of PCs can be found in Figure S6 (SM).

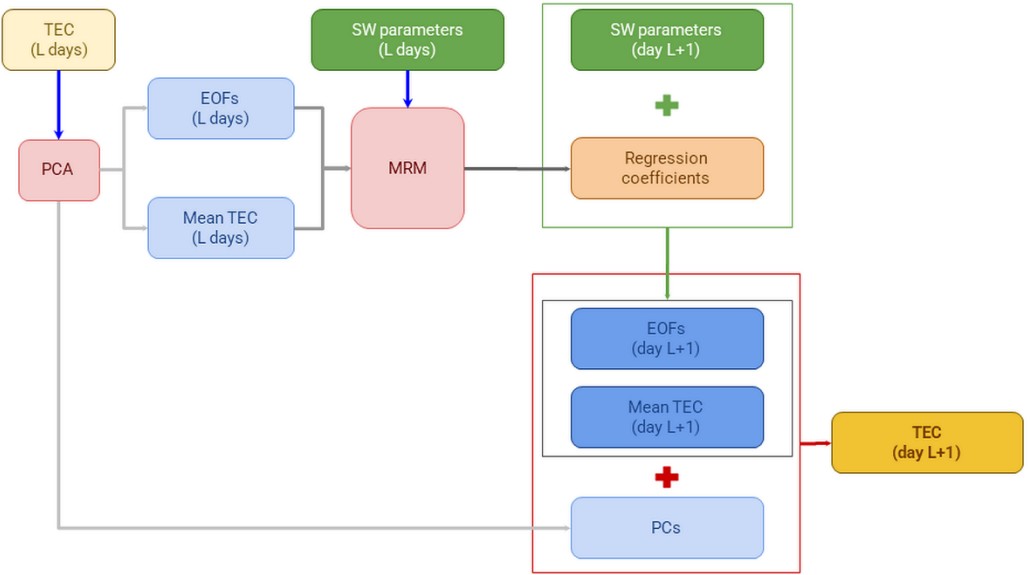

**Figure 1.** The PCA-MRM TEC model scheme.

The amplitude of each of these modes (EOFs) varies from day to day. During the second stage, these EOFs as well as the 1 d mean TEC series (all of 1 d time resolution) are submitted to the regression analysis to construct MRMs that make correspondences between these TEC parameters and the variations of SWp selected as predictors. Every time a "best subset" of predictors is estimated to maximise the $R_{adj}^2$ of the resulting regression model. MRMs are constructed using lags of 1 or 2 days between the variations of SWp and TEC (SWp lead).

As a result, the regression coefficients are generated allowing to use them at the third and final stage to reconstruct (forecast) TEC for the following day, day *L* + 1: we use correspondingly lagged SWp series as predictors to forecast the daily mean TEC, EOF1 and EOF2 values for that day and combining them with PC1 and PC2 to reconstruct (forecast) the 1 h TEC series for the day *L* + 1. No negative 1 d mean TEC and EOF1 series were allowed: in case MRMs forecast negative values of 1 d mean TEC or EOF1 they were multiplied by −1. The PCA-MRM models are denoted as PCA-MRM (*L*##, lag#) where *L*## is the length of the input time interval in days and lag# is the lag between the TEC and space weather parameters in days.

The interim 1 h TEC series forecasts were made separately for the MRM models with lags of 1 and 2 days, and the final forecast is constructed as the arithmetic mean of these forecasts, hereafter PCA-MRM (*L*##, mean.lag1.2).

To build and validate our model, we used the TEC data observed between January 1 and December 31 of 2015 in Lisbon airport and the SWp series for the same time interval. The assessment of the PCA-MRM model forecasting quality is presented in Section 5.

*4.2. Length of the Input Dataset L*

The length *L* of the input data series, both TEC and SWp, strongly affects the model performance. The shorter length may result in better representation of the TEC daily modes but will not be sufficient for the construction of reliable MRMs with so many regressors. On the other hand, larger *L* will allow constraining the regression coefficients well, but the TEC daily modes may be resolved with lower quality because of the seasonal changes of the TEC daily variation. We made tests for the PCA-MRM performance varying the *L* parameter from 25 to 45 days comparing the observed and forecasted series of 1 h TEC, 1 d mean TEC and the daily maximum (1 d max) TEC using metrics listed in Section 3.3 (some examples can be found in SM, Figures S7 and S8).

Overall, the lowest and the highest *L* values performed badly, and the best results were obtained for *L* in the interval from 28 to 33 days. Neither of the models has best performance with all metrics considered, however the models with *L* equal to 31 or 32 days seem to have overall better scores. Therefore, all further models were constructed using *L* = 31 and *L* = 32.

Another feature that was derived from these preliminary tests is that the models constructed as the arithmetic mean of the forecasts with lags of 1 and 2 days (PCA-MRM (*L##*, mean.lag1.2) very often perform better than the PCA-MRM (*L##*, lag#) models (see Figures S7 and S8 in SM). That is why we adopted the (PCA-MRM (*L##*, mean.lag1.2) approach for the final model.

## 5. Performance of the PCA-MRM TEC Model

The forecasting quality of the models was studied on the hourly (1 h TEC series) and the daily (1 d mean and 1 d max TEC series) time scales, during quiet (no solar flares, no geomagnetic disturbances) days, days with solar flares and days with geomagnetic disturbances, during different months, and during different hours of a day.

*5.1. General Performance*

Figures 2 and 3 show the observed (black lines) and forecasted using the PCA-MRM (L31, mean.lag1.2) model (red lines) 1 d mean and 1 d max TEC series, respectively. Figures 4 and 5 show examples of the forecast for the 1 h TEC series made with the PCA-MRM (L31, mean.lag1.2) model for June and February of 2015, respectively, the months with the worst and the best forecasts for the 1 h TEC series, respectively. The plots for all months of 2015 (including those for February and June) can be found in SM, Figures S9–S19. The results for the PCA-MRM (L32, mean.lag1.2) model are very similar and not shown.

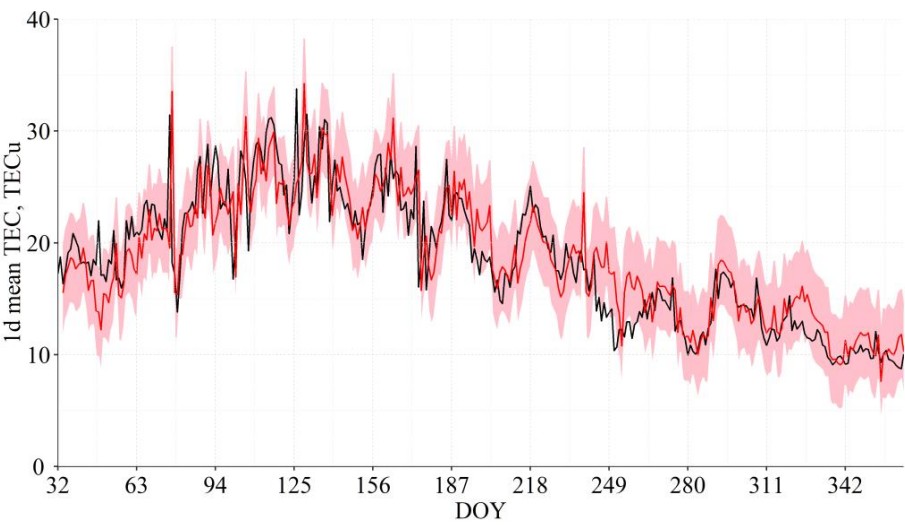

**Figure 2.** Observed (black) and forecasted using the PCA-MRM (L31, mean.lag1.2) model (red) 1 d mean TEC series for February–December of 2015; pink area shows 90% confidence interval (see Section 5.3).

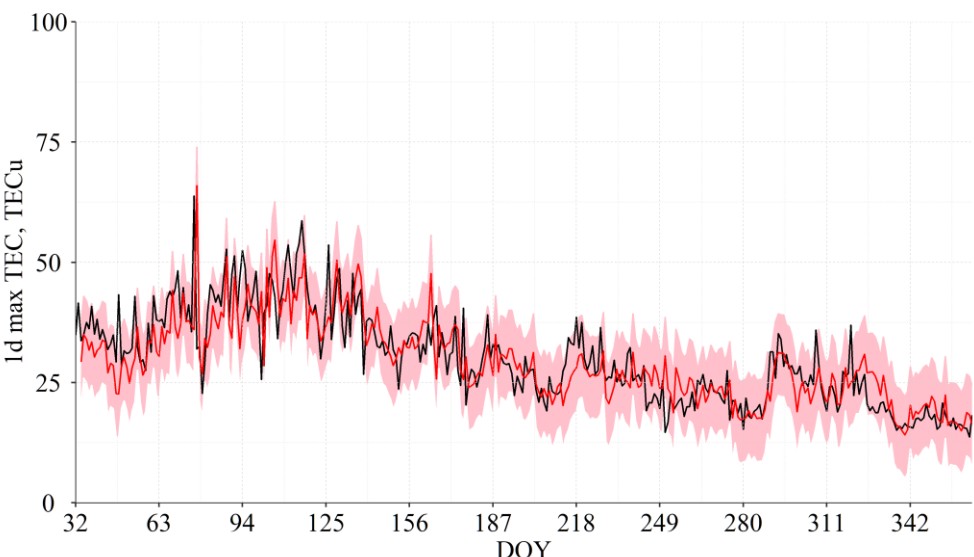

**Figure 3.** Observed (black) and forecasted using the PCA-MRM (L31, mean.lag1.2) model (red) 1 d max TEC series for February–December of 2015; pink area shows 90% confidence interval (see Section 5.3).

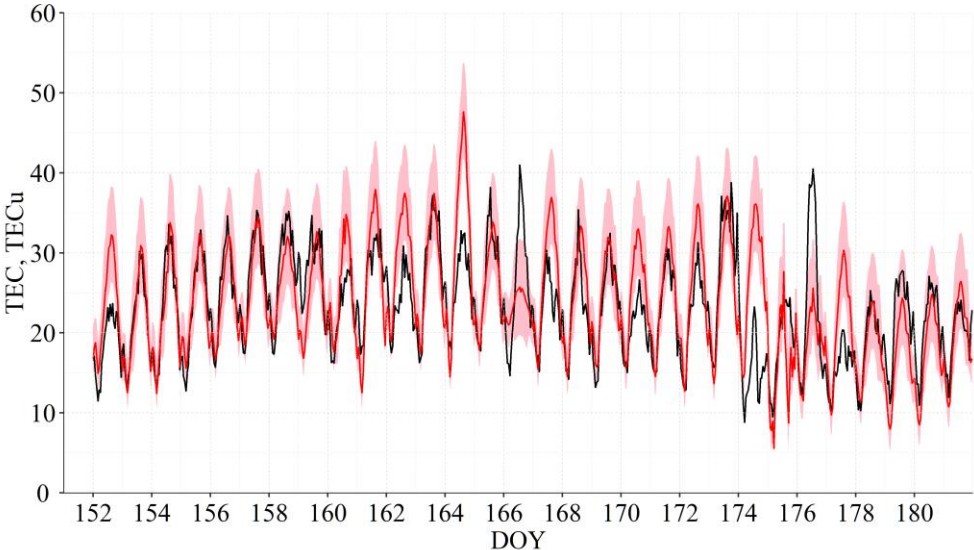

**Figure 4.** Observed (black) and forecasted using the PCA-MRM (L31, mean.lag1.2) model (red) 1 h TEC series for June 2015; pink area shows 90% confidence interval (larger version can be found in SM, Figure S13).

A simple visual analysis shows that the biggest differences between the observed and forecasted TEC series are seen during the days of geomagnetic storms: most prominently this is seen for March (Figure S10) and June (Figure 4 and Figure S13)—the months of the strongest storms of the solar cycle 24.

Table 1 shows the mean and the range values for the 1 h, 1 d mean and 1 d max observed and forecasted TEC series. As one can see, both the PCA-MRM (L31, mean.lag1.2) and PCA-MRM (L32, mean.lag1.2) models fit the observations similarly well but the performance of the PCA-MRM (L31, mean.lag1.2) model is slightly better. The scores (metrics values) of the two PCA-MRM models are shown in Table 2. Again, both models perform similarly well but the model with *L* = 31 days performs slightly better.

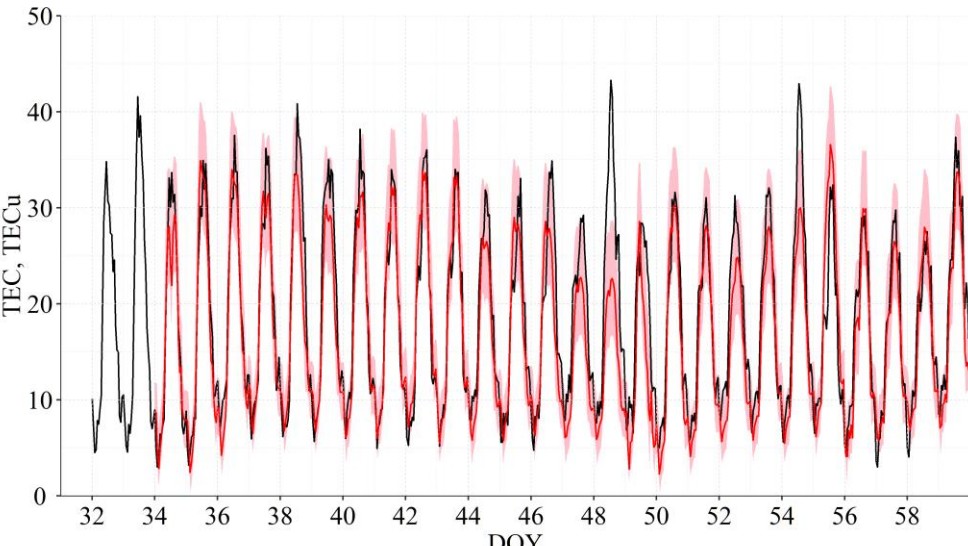

**Figure 5.** Observed (black) and forecasted using the PCA-MRM (L31, mean.lag1.2) model (red) 1 h TEC series for February 2015; pink area shows 90% confidence interval (larger version can be found in SM, Figure S9).

**Table 1.** The mean values and the range (in TECu) for different TEC parameters for the observed and forecasted series: 1 h TEC series, 1 d mean and 1 d maximum TEC series. Metrics of the models that are most close to the metrics of the observed TEC series are in bold.

| TEC Series | Parameter | Mean | Range |
|---|---|---|---|
| observed | 1 h TEC | 18.3 | 1.8/63.8 |
| | 1 d mean TEC | 18.3 | 8.7/33.8 |
| | 1 d max TEC | 30.0 | 13.7/63.8 |
| PCA-MRM (L31, mean.lag1.2) | 1 h TEC | **18.8** | **0.6/65.9** |
| | 1 d mean TEC | 18.8 | 7.6/**34.2** |
| | 1 d max TEC | **29.4** | **14.1/65.9** |
| PCA-MRM (L32, mean.lag1.2) | 1 h TEC | 17.7 | 0.5/68.7 |
| | 1 d mean TEC | **18.7** | **8.6**/34.8 |
| | 1 d max TEC | 29.2 | **14.1**/68.7 |

**Table 2.** Performance of the PCA-MRM (L##, mean.lag1.2) and the naïve models for the TEC series. Best scores of the models for each of the studied TEC series are in bold (1 h—columns 1 and 4, 1 d mean—columns 2, 5 and 7, and 1 d max—columns 3 and 6).

| Metric | PCA-MRM (L31, Mean.lag1.2) | | | PCA-MRM (L32, Mean.lag1.2) | | | Naïve Model |
|---|---|---|---|---|---|---|---|
| | 1 h | 1 d Mean | 1 d Max | 1 h | 1 d Mean | 1 d Max | 1 d Mean |
| minimum ΔTEC, TECu | **−14.7** | −11.9 | −27.8 | **−14.7** | −11.9 | **−25.8** | −11.4 |
| maximum ΔTEC, TECu | 10.7 | 15.3 | 34.0 | **6.7** | 16.6 | **26.7** | 7.9 |
| median ΔTEC, TECu | 0.27 | 0.23 | **−0.35** | 0.22 | 0.18 | −0.45 | 0.64 |
| mean ΔTEC (ME), TECu | **2.1** | 0.22 | −0.67 | 2.2 | **0.15** | −0.83 | 0.25 |
| $r$ | **0.89** | **0.88** | **0.80** | 0.88 | **0.88** | 0.79 | 0.86 |
| ExpV | **0.78** | **0.77** | **0.63** | 0.77 | **0.77** | 0.62 | 0.74 |
| $R^2$ | **0.80** | **0.77** | **0.63** | 0.79 | **0.77** | 0.61 | 0.73 |
| MAE, TECu | 1.80 | **2.0** | **4.2** | **1.76** | **2.0** | 4.3 | 2.5 |
| MaxE, TECu | **44.7** | 15.3 | 34.0 | 45.4 | 16.6 | **26.7** | **11.4** |
| RMSE, TECu | **4.27** | **2.8** | **5.9** | 4.32 | 2.9 | 6.0 | 3.0 |

It also seems that the ability of the PCA-MRM models to forecast the 1 d mean TEC values are better than for the 1 h series: as shown in Table 2, the values of the mean and median ΔTEC, MaxE and RMSE values are about 1.5–2 times smaller for the 1 d mean TEC series than for the 1 h TEC series.

Another standard way to assess a new model quality is to compare its forecasts to predictions made by a so-called naïve model, a model in which minimum manipulation of data is used to prepare a forecast [29]. One of the widely used naïve models simply uses the mean value of the studied parameter for a certain time interval as a forecasted value. In Table 2 we present a comparison of the PCA-MRM models to the naïve model calculated as averaging of the 1 d mean TEC series for the previous 31 days to calculate corresponding forecasts. As one can see, the PCA-MRM models outperform the naïve model except for the MaxE, max ΔTEC and min ΔTEC metric. The naïve model constructed in the same way for the 1 h TEC series (not shown), in general, forecasts lower TEC values than the observed one for the daytime and higher than the observed TEC values for the night-time, and the metric values are worse than those for the PCA-MRM models.

*5.2. Seasonal Variations of the Model's Performance*

The correlation between the observed and forecasted 1 h TEC series is generally high: $r = 0.72$–$0.92$, depending on a month, with the worst correlation obtained for June (Figures 4 and S13) and the best correlation obtained for February ($r = 0.92$, Figures 5 and S9) and October of 2015 ($r = 0.91$, Figure S17)—see Table 3. The differences between the observed and forecasted 1 h TEC series changes from day to day: in July and December MAE < 10 TECu for all days of a month; in February, March, May, September, October and November there were 1–2 days per month with 10 TECu < MAE < 15 TECu; in April, June and August there were 3–6 days per month with 10 TECu < MAE < 15 TECu; in February and April there was one day per month and in March there were 2 days with MAE ≈ 20 TECu.

**Table 3.** Correlation between the PCA-MRM (L31, mean.lag1.2) forecasted and the observed TEC values for different months.

| | **Months** | | | | | | | | | | |
|---|---|---|---|---|---|---|---|---|---|---|---|
| | **2** | **3** | **4** | **5** | **6** | **7** | **8** | **9** | **10** | **11** | **12** |
| $r$ | 0.92 | 0.83 | 0.86 | 0.83 | 0.72 | 0.83 | 0.81 | 0.80 | 0.91 | 0.87 | 0.87 |
| *p* values | <0.01 | <0.01 | <0.01 | <0.01 | <0.01 | <0.01 | <0.01 | <0.01 | <0.01 | <0.01 | <0.01 |

To estimate the quality of the PCA-MRM forecasts on the monthly time scale the monthly means were calculated for SWp (both those that were used as predictors and others) and for the TEC parameters: 1 d mean and 1 d max TEC and ΔTEC values (both observed and forecasted by the PCA-MRM models), and the 1 h TEC forecasted series (Figure 6). The monthly means of the 1 d mean and 1 d max ΔTEC values show strong anti-correlation with the monthly means of the solar UV (F10.7 and Mg II) and XR fluxes. This anti-correlation results from the underestimation of the amplitude of the TEC daily variations during time intervals with high levels of the solar UV and XR irradiance. This underestimation of the TEC daily mean and max values, at least partly, is connected to the underestimation of the flares' effect on the daily TEC values, but corresponding correlation coefficients between ΔTEC and the number of the solar flares (Figure 6) are low in the absolute values and statistically non-significant.

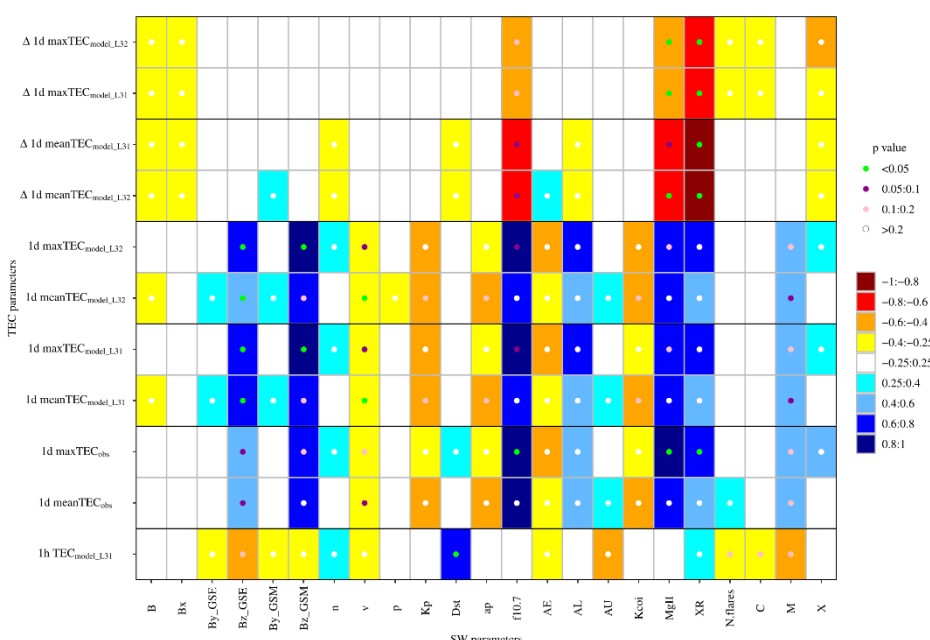

**Figure 6.** Correlation coefficients between TEC parameters and SWp on the monthly time scale. TEC parameters (*Y*-axis bottom up): forecasted TEC 1 h, forecasted and observed 1 d mean and 1 d max series, 1 d mean and 1 d max ΔTEC series; SWps are along the *X*-axis; *r* values are shown as coloured tiles and *p* values are shown as coloured dots.

Figure 6 allows the comparison of the behaviour of the modelled and observed TEC series with the variations of SWp—compare lines 2 and 3 from the bottom (1 d mean and 1 d max observed TEC, respectively) with lines 4–5 and 6–7 for the PCA-MRM (L31, mean.lag1.2) and PCA-MRM (L32, mean.lag1.2) models, respectively. The TEC 1 d series produced by both models show correlations similar to ones of the observed TEC for such SWp as Bz and n, F10.7 and Mg II, solar XR flux, the number of M flares, and geomagnetic AL and AU indices (all TEC series correlate with SWp); or v, Kp and $K_{COI}$, ap, and AE (all TEC series anti-correlate with SWp). The correlations between TEC and SWp series that are obtained only for the modelled series (e.g., with B, By and p) or seen only for the observed 1 d TEC series (with Dst and the number of all flares N) are low ($|r| < 0.4$) and statistically non-significant (*p* value > 0.2). Thus, the PCA-MRM models generate TEC series that maintain similar behaviour in respect to space weather variations.

Finally, the analysis of the correlation coefficients between the observed and forecasted 1 h TEC series calculated for individual months (Table 3) in dependence on the mean level of the solar, interplanetary and geomagnetic activity level (see Figure 6, bottom row and Figures S1–S5 in SM) shows that the PCA-MRM models provide best forecasts for months with lowest disturbance level: (1) the months with highest Dst values (Dst ≈ 0 nT), which are the months with lowest geomagnetic activity levels, and/or (2) the months with low number of flares. It is interesting to note that the correlation coefficients obtained for March and October of 2015, months with strong geomagnetic storms (see also [20]), are quite high ($r = 0.83$ and 0.91, respectively), but for June 2015, another month with a strong storm, the correlation is low ($r = 0.72$). Thus, it seems that what is more important for the PCA-MRM model's performance is not short-living events such as storms but the general level of the solar and geomagnetic activity during those *L* days that are used to build a forecast for a particular day.

*5.3. General Performance of the Hourly Time Scale*

On the hourly time scale the ΔTEC values forecasted by the PCA-MRM (L31, mean.lag1.2) model are in the range from −27.9 TECu to 44.7 TEC for individual days with mean ME = 2.1 TECu. The MAE values are in the range from 0 TECu to 44.7 TEC with mean MAE = 1.8 TECu.

The daytime hours were defined as the hours from 09:00 UTC to 19:00 UTC and the night-time hours were defined as the hours from 00:00 UTC to 07:00 UTC and from 21:00 UTC to 23:00 UTC of each day independently of the month. Two gaps of 2 h between the day- and night-hours subsets were chosen to avoid the influence of the seasonal variation of the sunrise and sunset times and, consequently, of the phase of the daily TEC variation.

For the whole number of the days with forecast by the PCA-MRM models, MAE = 3.9 TECu for the day hours (ME = 0.5 TECu), and MAE = 2 TECu for the night hours (ME = −0.1 TECu) or twice smaller than for the daytime.

Table 4 shows metrics of the observed and forecasted, PCA-MRM (L31, mean.lag1.2), TEC series calculated for all forecasted days for all hours and separately for the day and night hours. For the PCA-MRM (L31, mean.lag1.2) model we have forecasts for 332 days of 2015, since the first 33 days of 2015, from 1 January to 2 February, are used to build the first possible forecast. As one can see, SDs of the observed and forecasted 1 h series calculated are very close. MAE is about 3–4 TECu with MaxE ~8 TECu. The errors and SD are lower for the night hours than for the day hours: MAE ~2 TECu vs. ~4 TECu and MaxE ~4.5 TECu vs. ~8 TECu, respectively.

**Table 4.** The standard deviations of the observed and forecasted TEC, and the standard deviation of ΔTEC and the mean errors for the PCA-MRM (L31, mean.lag1.2) model: all hours, day hours and night hours.

| Parameter | All Hours | Day Hours | Night Hours |
|---|---|---|---|
| $SD_{TEC\ obs}$, TECu | 6.45 ± 0.15 | 3.9 ± 0.10 | 2.50 ± 0.06 |
| $SD_{TEC\ forec}$, TECu | 6.54 ± 0.14 | 3.6 ± 0.10 | 2.20 ± 0.06 |
| $SD_{\Delta TEC}$, TECu | 2.90 ± 0.09 | 2.9 ± 0.09 | 1.78 ± 0.05 |
| MAE, TECu | 2.90 ± 0.10 | 3.9 ± 0.16 | 2.00 ± 0.07 |
| RMSE, TECu | 3.70 ± 0.12 | 4.5 ± 0.17 | 2.40 ± 0.07 |
| ME, TECu | 0.20 ± 0.15 | 0.5 ± 0.24 | −0.10 ± 0.11 |
| MaxE, TECu | 8.20 ± 0.25 | 7.9 ± 0.25 | 4.50 ± 0.14 |

Table 5 presents the daily $R^2$ and ExpV scores for the PCA-MRM (L31, mean.lag1.2) model: it shows the percentage of days with $R^2$ and ExpV above a certain threshold. The scores are calculated for each day for all hours and for the day/night hours separately. Please note that $R^2$ or ExpV can be negative for days with an extremely bad performance of a model when the MSE or $\sigma^2_{\Delta y}$, respectively, are larger than $\sigma_y{}^2$ (see Equations (1), (2) and (6)).

**Table 5.** Percentage of the forecasted days (out of 332) with $R^2$ and ExpV scores in a certain range (PCA-MRM (L31, mean.lag1.2) model).

| Parameter | Time | Threshold | | |
|---|---|---|---|---|
| | | ≥0 | ≥0.5 | ≥0.8 |
| $R^2$ | all hours | 88.0% | 74.7% | 41.9% |
| | day hours | 49.4% | 26.0% | 5.00% |
| | night hours | 57.0% | 29.0% | 7.00% |
| ExpV | all hours | 95.5% | 88.3% | 58.0% |
| | day hours | 76.5% | 48.5% | 28.3% |
| | night hours | 80.0% | 46.4% | 14.8% |

Tables 6–8 show the mean and median values of different error scores (daily mean and median MAE and RMSE, ME, and MaxE) as well as the percent of days with the scores in a certain range. The last column shows the range of the scores observed for 90% of all forecasted days. The scores are calculated for each day for the whole day and for the day/night hours only. As one can see, on average, the absolute value of the forecast error is ~3 TECu with an error of ~2 TECu for the night hours. For all hours, 80–90% of the days

have errors with the absolute values not exceeding 5 TECu, whereas for the day hours only ~70% of the days have errors with the absolute values in this range, and almost for all days (95%) the absolute values of the errors for the night hours do not exceed 5 TECu. For 90% of the forecasted days MAE is in the range 0/5 TEC for all hours, 0/7 TECu for the day hours and 0/4 TECu for the night hours. The mean and median values of ME allows us to assume that for the day hours the PCA-MRM models tend to overestimate TEC values (ME > 0), whereas for the night hours there is a tendency to underestimate (ME < 0).

**Table 6.** Mean and median values and the percentage in a certain range for the MAE and RMSE scores for the PCA-MRM (L31, mean.lag1.2) model.

| Parameter | Time | Mean Value, TECu | Median Value, TECu | Range, TECu | | | For 90% of the Days a Metric Is in the Range, TECu |
|---|---|---|---|---|---|---|---|
| | | | | 0/3 | 0/4 | 0/5 | |
| MAE | all hours | 2.9 | 2.6 | 61.7% | 80.0% | 91.5% | 0/4.7 |
| | day hours | 3.9 | 3.2 | 46.0% | 63.0% | 77.7% | 0/6.5 |
| | night hours | 2 | 1.7 | 80.0% | 93.0% | 95.0% | 0/3.7 |
| RMSE | all hours | 3.7 | 3.3 | 43.7% | 65.7% | 82.2% | 0/5.6 |
| | day hours | 4.5 | 2.7 | 32.5% | 54.0% | 67.8% | 0/7.3 |
| | night hours | 2.4 | 2.1 | 73.2% | 86.0% | 95.0% | 0/4.3 |

**Table 7.** Mean and median values and the percentage in a certain range for the MAE and RMSE scores for ME(L31, mean.lag1.2) model.

| Parameter | Time | Mean Value, TECu | Median Value, TECu | Range, TECu | | | For 90% of the Days a Metric Is in the Range, TECu |
|---|---|---|---|---|---|---|---|
| | | | | −1/1 | −2/2 | −3/3 | |
| ME | all hours | 0.2 | 0.13 | 36.7% | 59.3% | 77.0% | −5.0/5.0 |
| | day hours | 0.5 | 0.58 | 31.6% | 48.0% | 62.0% | −6.0/6.0 |
| | night hours | −0.1 | −0.03 | 58.0% | 77.0% | 85.0% | −3.5/3.5 |

**Table 8.** Mean and median values and the percentage in a certain range for the MAE and RMSE scores for the MaxE(L31, mean.lag1.2) model.

| Parameter | Time | Mean Value, TECu | Median Value, TECu | Range, TECu | | | For 90% of the Days Metric Is in the Range, TECu |
|---|---|---|---|---|---|---|---|
| | | | | <1 | 1/5 | 1/10 | |
| MaxE | all hours | 8.2 | 7.2 | 0.30% | 22.0% | 75.0% | 1/13.5 |
| | day hours | 7.9 | 7 | 0.30% | 26.5% | 77.7% | 1/13.5 |
| | night hours | 4.5 | 4 | 0.90% | 65.0% | 97.0% | 1/8.00 |

MaxE is lower than 5 TECu for ~25% of the day hours and 65% of the night hours, and for 90% of the days it is lower than 13.5 TECu for the daytime and 8 TECu for the night-time.

For the 1 h TEC series 90% of the forecasted values have ΔTEC in the range ± 6 TECu, 90% of the forecasted 1 d max values have ΔTEC in the range from −9 to 8 TECu, and 90% of the forecasted 1 d mean values have ΔTEC in the range ±4 TECu. Thus, we estimate the 90% confidence intervals for 1 d mean TEC series as ±4 TECu, for the 1 d max TEC series as ±8.5 TECu, and for the 1 h TEC series they are ±6 TECu for the day hours and ±3 TECu for the night hours. Figure 4, Figure 5 and Figure S9–S19 in SM show PCA-MRM (L31, mean.lag1.2) model forecasts with 90% confidence interval (pink area) for the 1 h TEC series for individual months from February to December and Figures 2 and 3 show similar forecasts for the 1 d mean and 1 d max TEC series, respectively. As one can see, the observed TEC series (black lines) fit very well into the 90% confidence intervals of the PCA-MRM model.

### 5.4. Assessment of the Forecasting Skills during Quiet Days

The disturbed days (DD) were defined as days with daily mean values of the geomagnetic indices above/below a certain threshold (daily mean Dst $\leq -40$ nT, and/or ap $\geq 40$, and/or the daily mean of Kp $\geq 4.5$) and/or with at least three solar flares of the C or M classes. All other days were considered as quiet days (QD).

In general, for QDs the daily ME during the daytime is between 0.6 and 0.95 TECu and during the night-time it is between $-0.02$ and 0.41 TECu; the MAE values are ~3.4 TECu for the daytime and are ~2 TECu during the night-time. Thus, during QDs the amplitude of MAE during the night hours is ~1.7 times smaller than during the day hours.

To study the effect of the slightly elevated solar or geomagnetic activity, we made the following QD subsets (see Table 9): the subsets from QD1 to QD3 show the effect of 1–2 solar flares on the day-night differences of $\Delta$TEC, and the subsets QD1 and from QD4 to QD7 show the effect of the elevated geomagnetic activity. As one can see from Table 9, the increase in the daily number of the C flares from 0 to 2 results in the decrease in the mean $\Delta$TEC values for the night hours but the mean MAE values remain constant. This means that the PCA-MRM models tend to underestimate TEC values for the night hours of QD with 1 or 2 flares of up to C-class. For the day hours no statistically significant difference in $\Delta$TEC or MAE is obtained. An increase in the geomagnetic activity during QD seems to result in the overestimation by the PCA-MRM models of the night TEC values: $\Delta$TEC increases, MAE does not change.

**Table 9.** Parameters of the subsets of QDs and the mean $\Delta$TEC (in TECu) values. Data are for the PCA-MRM model with $L = 31$ days and averaging the model with lag = 1 day and lag = 2 days.

| Subset Number | $\Sigma$ Kp | ap | Dst, nT | Number of C Flares | Number of Days in the Subset | Day | | Night | |
|---|---|---|---|---|---|---|---|---|---|
| | | | | | | $\Delta$TEC | \|$\Delta$TEC\| | $\Delta$TEC | \|$\Delta$TEC\| |
| QD1 | <30 | <20 | >−40 | 0 | 51 | 0.95 ± 0.17 | 3.22 ± 0.11 | 0.32 ± 0.11 | 1.95 ± 0.07 |
| QD2 | <30 | <20 | >−40 | 0–1 | 113 | 0.64 ± 0.12 | 3.3 ± 0.08 | 0.07 ± 0.07 | 1.93 ± 0.05 |
| QD3 | <30 | <20 | >−40 | 0–2 | 144 | 0.8 ± 0.11 | 3.3 ± 0.07 | −0.02 ± 0.06 | 1.9 ± 0.04 |
| QD4 | <20 | <10 | >−40 | 0 | 37 | 0.93 ± 0.2 | 3.3 ± 0.12 | 0.25 ± 0.13 | 2.06 ± 0.09 |
| QD5 | <40 | <30 | >−50 | 0 | 59 | 0.81 ± 0.18 | 3.5 ± 0.13 | 0.33 ± 0.1 | 1.96 ± 0.07 |
| QD6 | <50 | <40 | >−60 | 0 | 61 | 0.87 ± 0.18 | 3.5 ± 0.12 | 0.37 ± 0.1 | 2 ± 0.07 |
| QD7 | <60 | <50 | >−70 | 0 | 64 | 0.91 ± 0.17 | 3.5 ± 0.12 | 0.41 ± 0.1 | 1.99 ± 0.07 |

The dependence of TEC variations on the weak geomagnetic activity seen even during the quiet days can be related to the coupling between the high- and mid-latitudinal regions of the ionosphere, as was pointed out in [30].

### 5.5. Assessment of the Forecasting Skills during Space Weather Events

In general, for DDs the daily ME values during the daytime are between –7.6 and 0.61 TECu and during the night-time they are between –0.73 and 0.54 TECu, the daily MAE values during the daytime are between 3.3 and 12.8 TECu and during the night-time they are ~2.2 TECu. Thus, during DDs the amplitude of MAE during the night hours is ~1.6–4.2 times smaller than during the day hours.

To study the effect of the solar flares and geomagnetic disturbances separately, we made the following DD subsets (Table 10): the subsets DD1 to DD3 show the effect of the solar flares on the day-night differences of $\Delta$TEC, and the subsets DD4 to DD6 show the effect of the geomagnetic activity (please note the low number of the days with geomagnetic disturbances but without the solar flares). The subset DD7–DD10 contains days either with at least one of the geomagnetic indices above (below for Dst) a threshold and with at least three solar flares of any type.

**Table 10.** Parameters of the subsets of DDs and the mean ΔTEC (in TECu) values. Data are for the PCA-MRM model with $L = 31$ days and averaging the model with lag = 1 day and lag = 2 days.

| Subset Number | Σ Kp | Ap | Dst | Number of Flares | Days in the Subset | Day | | Night | |
|---|---|---|---|---|---|---|---|---|---|
| | | | | | | ΔTEC | \|ΔTEC\| | ΔTEC | \|ΔTEC\| |
| | | | | no geomagnetic disturbance but solar flares | | | | | |
| DD1 | <30 | <20 | >−40 | >2 | 86 | −0.04 ± 0.16 | 3.8 ± 0.1 | −0.53 ± 0.09 | 2.07 ± 0.06 |
| DD2 | <30 | <20 | >−40 | >3 | 62 | 0.09 ± 0.19 | 3.6 ± 0.12 | −0.5 ± 0.11 | 2 ± 0.07 |
| DD3 | <30 | <20 | >−40 | >4 | 38 | −0.44 ± 0.21 | 3.3 ± 0.14 | −0.69 ± 0.13 | 2 ± 0.09 |
| | | | | geomagnetic disturbance without solar flares | | | | | |
| DD4 | >30 | >20 | <−40 | <1 | 8 | −0.48 ± 0.92 | 5.8 ± 0.69 | 0.54 ±0.31 | 2.1 ± 0.22 |
| DD5 | >40 | >30 | <−50 | <1 | 4 | −3.3 ± 1.6 | 7.7 ± 1.2 | −0.24 ± 0.47 | 2 ± 0.36 |
| DD6 | >50 | >40 | <−60 | <1 | 2 | −7.6 ± 3 | 12.8 ± 1.9 | −0.73 ± 0.9 | 3 ± 0.63 |
| | | | | geomagnetic disturbance and/or solar flares | | | | | |
| DD7 | >40 | >30 | <−50 | >2 | 161 | 0.36 ± 0.15 | 4.4 ± 0.11 | −0.14 ± 0.07 | 2.1 ± 0.05 |
| DD8 | >50 | >40 | <−60 | >2 | 155 | 0.37 ± 0.15 | 4.4 ± 0.11 | −0.13 ± 0.07 | 2.1 ± 0.05 |
| DD9 | >50 | >40 | <−60 | >3 | 121 | 0.53 ± 0.18 | 4.5 ± 0.13 | −0.11 ± 0.08 | 2.1 ± 0.06 |
| DD10 | >50 | >60 | <−100 | >3 | 112 | 0.61 ± 0.19 | 4.43 ± 0.14 | −0.16 ± 0.08 | 2.1 ± 0.06 |

As one can see from Table 10, the increase in the daily number of the flares above two per day on a geomagnetically quiet background (DD1–DD3) does not result in a systematic under- or over-estimation by the PCA-MRM models of the daytime ΔTEC ≈ 0 TECu (MAE ≈ 3 TECu), but the night-time ΔTEC are underestimated by ~0.5 TECu (MAE ≈ 2 TECu). In turn, the geomagnetic disturbances that are not accompanied by solar flares (DD4–DD6) seem to result in the underestimation by the PCA-MRM models of both the day and night ΔTEC values (ΔTEC tends to be negative). The amplitude of the differences between the forecast and the observations (|ΔTEC|) increases with the strength of a disturbance mostly for the day hours. Still, these conclusions are made on the low number of the days. Finally, when either geomagnetic disturbance (seen at least in one of the indices) or more than two solar flares are observed during a studied day (DD7–DD10), the daytime ΔTEC is overestimated and the night-time ΔTEC is underestimated by the PCA-MRM models with the MAE ≈ 4 TECu for the day hours and ~2 TECu for the night hours.

For all days of 2015 there is either very weak or no correlation between ΔTEC and the geomagnetic indices: for Dst $r = −0.2$ (*p* value < 0.01), for ap and the daily sum of Kp $r = 0$. On the other hand, if we compared the geomagnetic indices to MAE, there is relatively weak but statistically significant dependence of |ΔTEC| values on the strength of geomagnetic disturbances: for Dst $r = −0.35$ (*p* value < 0.01), for ap $r = 0.32$ (*p* value < 0.01) and for Kp $r = 0.25$ (*p* value < 0.01). Thus, for the geomagnetically disturbed days the forecasting quality of the PCA-MRM models decreases, MAE increases, but the under- and over-estimations of the daily mean TEC are observed with more or less similar frequencies.

For the geomagnetically disturbed days, the dependence of ΔTEC and |ΔTEC| on the geomagnetic activity is different (see Table 11), however not all correlations are statistically significant due to the low number of corresponding events. For Dst the correlation coefficients obtained for |ΔTEC| is higher in the absolute values than the correlation coefficients obtained for ΔTEC showing, again, that the over- and under-estimations of the daily mean TEC are observed with similar frequencies.

**Table 11.** Correlation coefficients *r* between ΔTEC and SW parameters observed during non-quiet days: days with larger number of flares or geomagnetic disturbances; *p* values are in parentheses. Statistically significant (95%) *r* are in bold.

| Days with/ | Correlation with/ | r for ΔTEC | r for \|ΔTEC\| |
|---|---|---|---|
| Daily mean Dst −40 nT | Dst | **−0.42 (0.03)** | **−0.58 (<0.01)** |
| Daily mean ap ≥ 40 | ap | **−0.46 (0.05)** | 0.36 (0.25) |
| Daily mean Kp ≥ 4.5 | Σ Kp | −0.44 (0.13) | 0.21 (0.55) |

The dependence of ΔTEC and |ΔTEC| on the number of the solar flares was studied separately for flares of different types (C and M flares and the total number of flares per day N). Please note that most of the flares observed in 2015 were of the C-type (951 out of 1018 flares observed in 2015). We found no correlation between ΔTEC or |ΔTEC| and the number of flares N when all events are considered, however, if we consider only days with at least five flares of any type, there is a weak correlation between the number of flares per day and the difference between the observed and forecasted TEC shown in Table 12. The correlation coefficients are statistically significant (>95%) and show that the amplitude of the forecast error (ΔTEC and |ΔTEC|) grows with the number of flares observed per day.

**Table 12.** Correlation coefficients *r* between ΔTEC and SW parameters observed during non-quiet days: days with larger number of flares or geomagnetic disturbances; *p* values are in parentheses. Statistically significant (95%) *r* are in bold.

| Days with/ | Correlation with/ | r for ΔTEC | r for \|ΔTEC\| |
|---|---|---|---|
| 5 or more flares of all type | Number of flares N | **0.26 (<0.01)** | **0.34 (<0.01)** |
| 5 or more of C flares | Number of C flares | **0.32 (<0.01)** | **0.36 (<0.01)** |
| 2 or more of M flares | Number of M flares | −0.37 (0.18) | −0.26 (0.28) |

The analysis of the ΔTEC and |ΔTEC| distribution shows that for the days with a moderate number of flares (<5) there is a tendency for the PCA-MRM models to underestimate the daily mean TEC values, whereas for the days with larger number of flares (5 or more) there is a tendency for the models to overestimate the daily mean TEC values.

The number of the M flares observed in 2015 (66 flares) is much lower than the number of the C flares. In most cases, there was only one flare of this class per day, and only for 13 days of 2015 two or more M-class flares were observed. Therefore, the results of the correlation analysis for these days (Table 12, bottom row) are statistically non-significant, and we cannot make a conclusion on the existence of the dependence of ΔTEC on the number of the flares of this type.

Since during 2015 there was only one short-living flare of the X type, no conclusion of the effect of this type of flares on the forecasting quality of the PCA-MRM model can be made.

## 6. Discussion and Conclusions

We propose a PCA-MRM model based on the principal component analysis (PCA) and the multiple linear regression (MRM) of the amplitudes of the PCA modes to forecast TEC. Several space weather parameters are used as regressors for MRMs. The analysis of the performance of this model on the data obtained in 2015 showed that such a model can be successfully used to forecast TEC variations at middle latitudes.

The best length of the input data set was found to be 31 or 32 days. The best performance is seen when the PCA-MRM forecasts made with time lags of 1 and 2 days (space weather parameters lead) are averaged. This, to our mind, reflects the fact that different space weather forcings (e.g., solar flares and geomagnetic disturbances) affect ionospheric conditions on different time scales. Averaging of the forecasts made with different time lags allows to combine effects of different forcings. This is in agreement with results of [4,12].

For the tested time interval (February to December 2015) and for a mid-latitudinal location (Lisbon, Portugal) the PCA-MRM model allows 90% confidence intervals of 6 TECu for day hours and 3 TECu for night hours, on average. These confidence intervals are calculated using all available days and do not take into account the level of the solar or geomagnetic activity.

For the quiet days (days without M or X flares and no more than 1 flare of the class C or below, and without geomagnetic disturbances) the MAE and RMSE are about 3–3.5 TECu; for geomagnetically disturbed days without flares MAE and RMSE are about 5–7 TECu; for the days with M or X flares and/or with more than 1 flares of the class C or below MAE

and RMSE are about 3.5–4 TECu. Thus, the PCA-MRM model performs well during days without significant geomagnetic disturbances even if a flare is observed. Solar flares do not significantly deteriorate the quality of the forecasts compared to the one for the quiet days.

The daily mean and monthly mean ME and MAE anti-correlate with the solar UV and XR flux: the PCA-MRM model systematically underestimates TEC values for days with high levels of the solar UV and XR irradiance. The daily mean and monthly mean ME and MAE depend on the mean values of the geomagnetic indices Dst, Kp, ap: the PCA-MRM model both under- and over- estimates TEC values during days with geomagnetic disturbances with approximately similar rates, however large overestimations are seen more often than large underestimations.

Compared to other TEC models, both global and regional, the PCA-MRM model presented here and based on a single-station TEC data performs very well having RMSE (and other metrics) in the same range as for other models (see Section 1 and [1–12]). For example, our model provides for the 1 h TEC series RMSE = 3.70 TECu for all available days of 2015, RMSE = 4.5 TECu for all days but day hours and RMSE = 2.40 TECu for all days but night hours only, whereas, for example, the models [2–13] provide RMSE in the range from 2 to 5 TECu for quiet/not separated days and RMSE up to 10–15 TECu for storm days. The PCA-MRM model's performance also deteriorates during geomagnetically active periods while solar flares alone have no strong effect on the model's performance. Besides, our results allow providing different confidence intervals for the day and night hours: the forecast errors for the night hours are ~1.6–2 times smaller than one for the day hours (depending on the level of geomagnetic activity). This is in an agreement with [12].

On the other hand, our model has at least two advantages compared to many other models. First, we do not need to make assumptions on the character of the daily and seasonal TEC variations since the amplitude and the phase of the daily TEC variation for a certain time interval (e.g., a month) is extracted automatically from the data used to build a model. Second, our model uses a limited set of the TEC and SWp observations: just 31 or 32 days of data (1 h data for TEC and 1 d data for SWp) are needed to build the model. Thus, the model is not demanding on the database length (computer memory) and computational time, and, therefore, can be used by small private enterprises (like those participating in the ESA Small ARTES Apps project SWAIR) that monitor ionosphere conditions for GNSS-service consumers as a simple model for a short-term (1 day ahead) regional TEC forecasting.

**Supplementary Materials:** The following are available online at https://www.mdpi.com/article/10.3390/atmos13020323/s1 in separate pdf files: Figures S1–S5: Variations of space weather parameters for 2015 (Figs_S1-S5.pdf); Figure S6: Examples of PC1s and PC2s of the TEC variations (Fig_S6.pdf); Figures S7 and S8: Scores of PCA-MRM models with different *L* values for 1 h TEC, 1 d mean and 1 d max TEC series, respectively (Figs_S7-S8.pdf); Figures S9–S19: Observed and forecasted using the PCA-MRM (L31, mean.lag1.2) model 1 h TEC series with 90% confidence interval for February to December of 2015 (Figs_S9-S19.pdf).

**Author Contributions:** Conceptualization, A.L.M., T.B. (Teresa Barata), T.B. (Tatiana Barlyaeva); methodology, A.L.M., T.B. (Tatiana Barlyaeva); software, A.L.M., T.B. (Tatiana Barlyaeva); validation, A.L.M., T.B. (Teresa Barata), T.B. (Tatiana Barlyaeva).; formal analysis, A.L.M., T.B. (Tatiana Barlyaeva); investigation, A.L.M.; resources, T.B. (Teresa Barata); data curation, A.L.M., T.B. (Tatiana Barlyaeva); writing—original draft preparation, A.L.M.; writing—review and editing, T.B. (Teresa Barata), T.B. (Tatiana Barlyaeva); visualisation, A.L.M.; project administration, T.B. (Teresa Barata); funding acquisition, T.B. (Teresa Barata) All authors have read and agreed to the published version of the manuscript.

**Funding:** This research was supported through the project "SWAIR—Space weather impact on GNSS service for Air Navigation," ESA Small ARTES Apps, https://about.swair.ptech.io/ (accessed on 14 February 2021) and the project "PRIME: Portuguese Regional Ionosphere Model " (EXPL/CTA-MET/0677/2021) funded by the Fundação para a Ciência e a Tecnologia (FCT). IA is supported by Fundação para a Ciência e a Tecnologia (FCT) through the research grants UIDB/04434/2020 and UIDP/04434/2020.

**Institutional Review Board Statement:** Not applicable.

**Informed Consent Statement:** Not applicable.

**Data Availability Statement:** The TEC data for 2015 are available at Barlyaeva, T.; Barata, T.; Morozova, A.; 2020. Datasets of ionospheric parameters provided by SCINDA GNSS receiver from Lisbon airport area, Mendeley Data, v1 http://dx.doi.org/10.17632/kkytn5d8yc.1 (accessed on 14 February 2021). "SCINDA-Iono" toolbox for MATLAB by T. Barlyaeva is available online at https://www.mathworks.com/matlabcentral/fileexchange/71784-scinda-iono_toolbox (accessed on 14 February 2021). We acknowledge the mission scientists and principal investigators who provided the data used in this research: The TEC data sets are from the Royal Observatory of Belgium (ROB) data base and are publicly available in IONEX format at ftp:/gnss.oma.be/gnss/products/IONEX/ (accessed on 14 February 2021), see also [21] for more information. The Dst index is from the Kyoto World Data Center http://wdc.kugi.kyoto-u.ac.jp/dst_final/index.html (accessed on 14 February 2021). Geomagnetic data measured by the Coimbra Geomagnetic Observatory are available at the World Data Centre for Geomagnetism web portal http://www.wdc.bgs.ac.uk/dataportal/ (accessed on 14 February 2021). The solar wind data and the ap index are from the SPDF OMNIWeb database. The OMNI data were obtained from the GSFC/SPDF OMNIWeb interface at https://omniweb.gsfc.nasa.gov (accessed on 14 February 2021), see also [31] for more details. The Mg II data are from Institute of Environmental Physics, University of Bremen http://www.iup.uni-bremen.de/gome/gomemgii.html (accessed on 14 February 2021), see also [23] for more information. The data on the variations of the solar XR flux are from the LASP Interactive Solar Irradiance Data Center (LISRD, http:/lasp.colorado.edu/lisird/, accessed on 14 February 2022). LISIRD provides a uniform access interface to a comprehensive set of Solar Spectral Irradiance (SSI) measurements and models from the soft X-ray (XUV) up to the near infrared (NIR), as well as Total Solar Irradiance (TSI). The XRTIMED data are from the Solar EUV Experiment (SEE) measures the solar ultraviolet full-disk irradiance for the NASA TIMED mission. Level 3 data represent daily averages and are filtered to remove flares available at http:/lasp.colorado.edu/lisird/data/timed_see_ssi_l3/ (accessed on 14 February 2021). The X-ray Flare dataset was prepared by and made available through the NOAA National Geophysical Data Center (NGDC). The data about the solar flares for 2015 are from https://www.ngdc.noaa.gov/stp/space-weather/solar-data/solar-features/solar-flares/x-rays/goes/xrs/goes-xrs-report_2015_modifiedreplacedmissingrows.txt (accessed on 14 February 2021).

**Acknowledgments:** We are grateful to SEGAL (Space and Earth Geodetic Analysis laboratory) and personally to Rui Fernandes from University of Beira Interior (Portugal) for the access to the SCINDA receiver data. We are grateful to the reviewers for useful comments and suggestions.

**Conflicts of Interest:** The authors declare no conflict of interest.

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
