# Peer review of "PCA-MRM Model to Forecast TEC at Middle Latitudes"

_atmosphere, doi:10.3390/atmos13020323_

Round 1
Reviewer 1 Report
- Presentation is very poor. Methodology is not described properly. Discussion is almost nil.
- Section 2.2: A figure showing all SWp during the study period is required for the reader to understand the remaining part of the paper, particularly sections 5.5 and 5.6.
- What is the scientific rationale behind making forecasts with 1 and 2 day lags and then averaging (lines 258-260)? The response of ionosphere to solar irradiance is almost negligible on hourly time scales. On the other hand, the response to storms etc. is much larger than 1-2 days.
- In lines 205-210, it is very unclear as to how the MRM was constructed. If I understand correctly, the cadence of the EOFs is 1 hr and that of the 1d TEC and SWp is 1 day. Howa re they combined? No references are given for any of the methods. The pioneering papers of PCA and multi-variable regression need to be cited. How is the 90% confidence interval calculated for figures 2-5 and supplementary figures?
- ‘so-called’ is not at all a phrase that is expected in a scientific paper. Use of appropriate language and references wherever necessary is required.
- Can R_adj^2 be elaborated?
- Line 284: There is no section 5.1
- Line 286 – Replace ‘1 mean’ with ‘1d mean’.
- There is almost no discussion of the results obtained in light of earlier works (although a few have been mentioned in the introduction section). Entire section 5 only contains ‘Results’. In what ways has the current model performed in comparison with earlier works needs to be discussed. The merits and demerits have to be highlighted. In what way is the new model adding to existing capabilities of TEC prediction?
- Table 1 and similar: Does ‘range’ imply min to max values? Could standard deviation not be used as a more statistical representation?
- Table 1 Caption: Define ‘most close’ scientifically in terms of statistical parameters. This is a very vague and unscientific description.
- Table 2: ‘r’ value in columns 4 and 5 is same, yet one is made bold and the other is not. Why?
- Line 318 – Give references to ‘naïve model’.
- Lines 351-358: An anti-correlation of monthly means of TEC with that of UV and XR irradiance is unexpected. If it is found, it needs to be discussed in more detail with sufficient reference to earlier works.
- Lines 376-377: Error in mean ME and MAE is significant at the fourth decimal value. This sems to be a little far-fetched.
- Tables 6 to 8 have different structures than Table 5. Captions need to be rewritten appropriately.
- Line 455: |Delta TEC| is not defined.
- Line 485: Is it ‘For all disturbed days of 2015…’?
- Lines 540-542 & Abstract: The model’s errors are greater than 3 TECu while errors in references [5,9,10,11,12, etc] are smaller. So, as asked above, in what way is the new model adding to existing capabilities of TEC prediction?
- Significant grammatical errors are present. A thorough proof reading is required.
- First time occurrences of acronyms are not defined (For eg., GNSS, GIMs). Sometimes, acronyms are defined multiple times (for eg., SD) and unnecessary acronyms are defined which are never used later (for eg., NOC).
- Figure captions are very poorly written. For eg., Figure 2 caption does not mention the month and year of the graph. Legends are missing in all figures.
- Quality of figures can be improved.
- The symbol ‘…’ is used throughout the paper multiple times and it is not recommended. To describe a range, ‘-‘ can be used.
- What are the units of PC1 and PC2 in Figure S1. If these are arbitrary units, how are they used in prediction of Actual TEC values?
- Where is month 1 in Figure S1? (It is understood that the prediction starts from Month2, but PC1 and PC2 should exist for month 1.) Legends can be renamed as Jan, Feb,….Dec instead of m.1, m.2… etc. Font size of tick marks and legend is very small and difficult to read.
- Figure S2: Correlation coefficients of lag 1.5 days is very high. Why? The figures are not explained adequately. The colour codes appear to have errors. For eg., bottom panels have lot of blue which does not exist in the colorbars. It needs to be verified. The panels need to be renamed as (a), (b), etc instead of (top left), (top right), etc. Also, Delta TEC has to be specified in the figure as to which error it pertains to. Similarly in Figure S3.
- Figure S4: Is this figure for March or February 2015?
- The titles of all supplementary figures are very crude and not aesthetic. Also, instead of month2, month3, use of names of the months is preferable.
- Figure S15 is important and should be moved to main paper. It has to be discussed in detail. The p value part is not discernible/comprehensible.
Overall Impression:
The work is a good attempt in developing a TEC prediction model for a regional station based on a combination of PCA and multi-variable regression methods. However, the presentation needs to be significantly improved. The manuscript is not acceptable in its present form and needs to be revised. Specific attention is required to elaborate the methodology and discussion sections. The authors have made quite a few careless mistakes in terms of writing and also did not give adequate thought to the final form of all figures. Each figure has to be self-explanatory (in terms of legends, etc.) and the caption should carry all details.
Author Response
Reply to Reviewer 1
We are very thankful to the Reviewer for the useful comments.
Please note that in the replies below we are referring to the numbering of the lines, figures and references in the revised paper (docx file with Markup).
Presentation is very poor. Methodology is not described properly. Discussion is almost nil.
We revised the text expanding the Methods and Discussion sections.
Section 2.2: A figure showing all SWp during the study period is required for the reader to understand the remaining part of the paper, particularly sections 5.5 and 5.6.
Corresponding figures are added to the Supplementary Materials (Figs. S1-S5).
What is the scientific rationale behind making forecasts with 1 and 2 day lags and then averaging (lines 258-260)? The response of ionosphere to solar irradiance is almost negligible on hourly time scales. On the other hand, the response to storms etc. is much larger than 1-2 days.
The concept of the PCA-MRM model we propose allows only for the lags between space weather parameters (SWp) and TEC series that are multiple of a day, not hour, thus we cannot predict short term events such as response of the ionosphere to flares of the forecasted day. However, both our previous studies (Morozova, A.L.; Barlyaeva, T.V.; Barata, T. Variations of TEC over Iberian Peninsula in 2015 due to geomagnetic storms and solar flares. Space Weather 2020, 18(11), p.e2020SW002516.) and some works cited in Sec. 1 show that lags of 1-2 days works fine for the TEC forecasting purposes. Multiple flares lead to increase of the solar UV and XR fluxes that lasts for more than 1 day and can affect TEC daily amplitudes. The geomagnetic storms we studied have significant effect on the TEC variations during the day of a storm and the following days, more prolonged effects are rare, at least at middle latitudes we take into consideration. We tested a possibility to add a 3-days lag but the performance of the model deteriorated, thus we rejected lags longer than 2 days. Finally, the proposed lags and, what is more important, there combination, allow the forecasting skills of the PCA-MRM TEC models to be of the same order as other TEC models of comparable concept and inputs.
In lines 205-210, it is very unclear as to how the MRM was constructed. If I understand correctly, the cadence of the EOFs is 1 hr and that of the 1d TEC and SWp is 1 day. How are they combined?
With PCA matrix made the way explained in the paper, the final PC series have 1h time resolution (24 values each) and EOF series has 1d time resolution (L values each, i.e., for the data presented in the paper they are 31 or 32 values long). Thus, there is no problem to combine them with SWp series which have, for this reason, 1d time resolution. A note is added to the revised text explaining this (ll. 206).
No references are given for any of the methods. The pioneering papers of PCA and multi-variable regression need to be cited.
Since PCA and linear regression are very widely used mathematical methods, we believe that lengthy explanations are not needed though we some references for PCA were given in the original text [refs. 24-26].
The linear regression seems for us even more well-known method to be explained in detail, but we added some extra sentences to the text (ll. 218-229). Since we invented no new algorithm to solve sets of linear regression equations but used the standard computation packages, we believe no detailed explanation is needed in our paper, but we added a reference to a math handbook [ref. 27] with detailed explanation of the methods and examples.
How is the 90% confidence interval calculated for figures 2-5 and supplementary figures?
The confidence intervals for the 1h, 1d mean and 1d max series are now explicitly mentioned in the text (ll. 488-491). They are calculated considering percentage of the numbers in Tab. 6 and ll. 486-488.
‘so-called’ is not at all a phrase that is expected in a scientific paper. Use of appropriate language and references wherever necessary is required. Can R_adj^2 be elaborated?
The “adjusted squared coefficient of multiple determination” or Radj2 is a standard parameter to estimate, e.g., the quality of the fitting. The differences between the “adjusted R2” and the ordinary R2 is that Radj2 is that the error sum of squares and total sums of square (see Eq. 3 in the text) are both divided by the number of degrees of freedom (a note is added to ll. 223-225). We used “so-called” just to introduce a term not yet mentioned in the text, however we deleted “so-called” from the revised text.
Line 284: There is no section 5.1
We apologise for this mistake. All sections are properly numbered now.
Line 286 – Replace ‘1 mean’ with ‘1d mean’.
Thank you for noting this typo.
There is almost no discussion of the results obtained in light of earlier works (although a few have been mentioned in the introduction section). Entire section 5 only contains ‘Results’. In what ways has the current model performed in comparison with earlier works needs to be discussed. The merits and demerits have to be highlighted. In what way is the new model adding to existing capabilities of TEC prediction?
We briefly described results presented in 13 papers, that we’re aware of, that in some way use PCA to forecast TEC, either in the form of maps or single-station series, ending section 1 with a short summary. The comparison of these previous results with our model as well as description of its advantages were already done in the original paper (Section 6), but we elaborated this section in the revised paper (now “6. Discussion and conclusions”).
Table 1 and similar: Does ‘range’ imply min to max values? Could standard deviation not be used as a more statistical representation?
Yes, “range” here is a parameter that gives min and max values. The standard deviation is also given for the analysed series (e.g., Tab. 4), however the range, to our mind, is also important, as well as the maximum error MaxE parameter, to give an idea of the lowest/highest observed and forecasted TEC values and errors. In particular, Tab. 1 allows reader to see that the forecasted series have very close ranges and there is no severe overestimation. (Please also see below the answer to another question about the range parameter and l. 240 in the revised text)
Table 1 Caption: Define ‘most close’ scientifically in terms of statistical parameters. This is a very vague and unscientific description.
Here, “most close” simply means that the value of a metric for a particular model is closer to the corresponding metric of the observed series: e.g., for the 1h TEC series:
mean TECobs = 18.3 TECu,
mean TECPCA-MRM L31 = 18.8 TECu,
mean TECPCA‑MRM L32 = 17.7 TECu.
Clearly, the PCA-MRM L31 model gives the mean TEC values closer to the observed ones, than the PCA-MRM L32 model and, therefore, the PCA-MRM L31 mode’s value is in bold in Table 1.
Table 2: ‘r’ value in columns 4 and 5 is same, yet one is made bold and the other is not. Why?
Because we compare metrics between the corresponding series: 1h with 1d series, 1d mean with 1d mean series, 1d max with 1d max series. Column 4 must be compared with column 1, while column 5 must be compared with columns 2 and 7. We modified the Table 2 caption to make it clear.
Line 318 – Give references to ‘naïve model’.
Done, [ref. 29]
Lines 351-358: An anti-correlation of monthly means of TEC with that of UV and XR irradiance is unexpected. If it is found, it needs to be discussed in more detail with sufficient reference to earlier works.
The anti-correlation with the solar UV & XR is found for the ΔTEC (errors of the model) averaged for 1 month time intervals, not for the monthly mean TEC series. It is discussed in ll. 384-396 and is related to “the underestimation of the amplitude of the TEC daily variations during time intervals with high level of the solar UV and XR irradiance”. We also added ll. 403-414 with discussion of other features of Fig. 6
Lines 376-377: Error in mean ME and MAE is significant at the fourth decimal value. This sems to be a little far-fetched.
Well, we agree that giving the standard error (SE) values with the 4th decimal value dose not correspond to the TEC measurements accuracy, still the SE values are very small. We removed SE values from the text.
Tables 6 to 8 have different structures than Table 5. Captions need to be rewritten appropriately.
Done
Line 455: |Delta TEC| is not defined.
It is defined by Eq. 4.
Line 485: Is it ‘For all disturbed days of 2015…’?
Yes, of course. As is mentioned in the text, e.g., ll. 129, ll. 283-284, 553 etc. We do not want to be too repetitive and mention this too many times through the paper.
Lines 540-542 & Abstract: The model’s errors are greater than 3 TECu while errors in references [5,9,10,11,12, etc] are smaller. So, as asked above, in what way is the new model adding to existing capabilities of TEC prediction?
The RMSE & MAE errors of our model, when calculated on the whole available data set are ~4 TECu for the daytime and ~2 TECu for the night-time, however the values used to calculate the 90% confidence intervals are slightly higher (6 and 3 TECu, respectively). This seems to be in line with RMSE/MAE of other models, some of them mentioned in Sec. 1 [refs. 2-13].
Please note that [ref. 9, 11] used only quiet time data while we make no separation, thus our error will be greater. The work [10] use only storm -time data and their RMSE are high (up to 10 TECu) with higher RMSE values for 2015, the year we use to train and test our model. The RMSE of [12] are not significantly better than ours. In general, even if RMSE/MAE of our model are larger than of other models, the difference does not exceed 1-2 TECu.
Also, the works [5,9-12] as well as other mentioned in Sec. 1, use, as a rule large data sets (either in time domain or both in time and space domains) which may results in lower error values, while our model use quite short input data sets and was tested on just 1 year of data. The direct comparison between all the models is complicated since they are not tested on the same TEC data set. We believe that for a model that do not need large computational resources (which is one of our goals) our model performs quite well (see Sec. 6).
Significant grammatical errors are present. A thorough proof reading is required.
The text was spellchecked, and grammatical error and misspelling are corrected to British English.
First time occurrences of acronyms are not defined (For eg., GNSS, GIMs). Sometimes, acronyms are defined multiple times (for eg., SD) and unnecessary acronyms are defined which are never used later (for eg., NOC).
GNSS is explained (l. 30), GIM & RIM were already introduced in the original text (ll. 58-58 and 48, respectively), duplicate explanations for SD and the NOC abbreviation are removed.
Figure captions are very poorly written. For eg., Figure 2 caption does not mention the month and year of the graph. Legends are missing in all figures.
We revised figure captions. Since the legends are not mandatory, they are added only to those plots where they are essential for understanding (like Fig. 6 or S6-S8). The plots with just 1-2 lines are left without the legend for clear view and explanations are done in the corresponding captions.
Quality of figures can be improved.
All figures are re-plotted aiming for better quality. Please also note that the figures for the final version will be used from the corresponding pdfs, not from the png files inserted in the text.
The symbol ‘…’ is used throughout the paper multiple times and it is not recommended. To describe a range, ‘-‘ can be used.
We now use “/” accordingly to the range new definition in ll. 240
What are the units of PC1 and PC2 in Figure S1. If these are arbitrary units, how are they used in prediction of Actual TEC values?
To our knowledge, PCs (as we define PCS and EOFs in Sec. 3.1 following e.g. [26]) are always considered to be in arbitrary units and EOFs have the units of the original series (TECu in our case). We added this note to the text (l. 206).
Where is month 1 in Figure S1? (It is understood that the prediction starts from Month2, but PC1 and PC2 should exist for month 1.) Legends can be renamed as Jan, Feb,….Dec instead of m.1, m.2… etc. Font size of tick marks and legend is very small and difficult to read.
Now all 12 series are shown; the legend has months names instead of numbers.
Figure S2: Correlation coefficients of lag 1.5 days is very high. Why? The figures are not explained adequately. The colour codes appear to have errors. For eg., bottom panels have lot of blue which does not exist in the colorbars. It needs to be verified. The panels need to be renamed as (a), (b), etc instead of (top left), (top right), etc. Also, Delta TEC has to be specified in the figure as to which error it pertains to. Similarly in Figure S3.
Figs. S7-S8 (old S2-S3) are replotted with better gradient scale and with new Y-axis labels. We prefer not to add labels to the panels to keep plots clearer, besides the named colour legends allows easy reading. Delta_TEC means MAE, as is said in the captions.
As to the higher correlations/lower errors of the averaged series (lag “mean1.2”), this is exactly why we applied the averaging of the forecasts with lags = 1 and 2 days to our model: the averaging of the forecasts with 2 lags improve the forecasting quality (see new text in ll. 302-306).
Figure S4: Is this figure for March or February 2015?
Corrected
The titles of all supplementary figures are very crude and not aesthetic. Also, instead of month2, month3, use of names of the months is preferable.
All figures are re-plotted aiming for better quality. Names of the months are provided in the captions.
Figure S15 is important and should be moved to main paper. It has to be discussed in detail. The p value part is not discernible/comprehensible.
Done. Discussed in ll. 384-414.

Reviewer 2 Report
This paper describes a forecast model using principal component analysis (PCA) and regression analysis to predict the TEC in the mid-latitude, especially to the geomagnetic disturbance and solar flare. The author first gives a detailed description of the previous studies on the empirical TEC model. And finally, they pointed out the potential disadvantages of these previous studies, which make their current studies look better. Then they introduce their method and data, including TEC data and different input parameters. After testing the performance of this TEC model under different conditions (quiet time, disturbed time, solar flare time, different months and UTs), they found that this model can well predict the TEC variations in around 30 days. This paper is organized and well written. I recommend the paper to have a minor revision before it is suitable for publication
Line 178 what is the temporal resolution of the OMNI data? 5-min or 1-hour?? Please clarify
Line 182 1-day resolution is too coarse for OMIN data such as IMF and solar wind, how can the daily average reflect the real geomagnetic condition??? While in Line 185, the author said 1h TEC series, which is in conflict with the description mentioned above, please double check
Line 437 Notice that Kp cannot be equal to 4.5, it can only be like 4 4.3 4.7. please change to Kp>4.5 or Kp>=4.3
In the section 5.5 about forecast during quiet days, the author shall also point out that the mid-latitude TEC is sensitive to geomagnetic disturbance, even during quiet time, compared to the response to lower atmospheric forcing. This may help explain the assessments results. The following paper can be a suitable reference to support the argument
Cai, X., Burns, A. G., Wang, W., Qian, L., Pedatella, N., Coster, A., et al. (2021). Variations in thermosphere composition and ionosphere total electron content under “geomagnetically quiet” conditions at solar-minimum. Geophysical Research Letters,
48, e2021GL093300. https://doi.org/10.1029/2021GL093300
Author Response
We are very thankful to the Reviewer for being so supportive of our work presented in this paper.

Reviewer 3 Report
This paper proposes an empirical model based on the principal component analysis (PCA) and the multiple linear regression (MRM) of the amplitudes of the PCA modes to forecast TEC. Many space weather parameters are used as regressors. The research background and the method are described unclearly. There are so many papers performing the ionosphere EOF empirical models, which makes this paper less innovative. What’s more, the reviewer has soem main concerns as follows.
The TEC data are from one GNSS station’s observation. How to derive the TEC from raw GNSS observation? Is the average of vertical TECs from different satellite with certain cutoff elevation? How to calibrate the TEC with ROB TEC data? These important information are very limited in the manuscript, so it can be not determine the TEC data quality.
So many space weather parameters are considered in the regression process. But the possible influences on the ionospheric variation are not provided in the manuscript. At least the physical connections should be given in the manuscript. Are the flare number with different levels necessary for the ionospheric forecasting? Due to the different response time of the ionosphere to solar flux and geomagnetic activity, the fixed delay time interval is not suitable for the regression process.
As the manuscript given: “All SWp series used in the PCA-MRM model have 1d time resolution”. I wonder what is the meaning of “mean lag1.2” , Is it the average TEC of the forecasting TEC with delay one day and with delay two day?
Author Response
Reply to Reviewer 3
We thank the Reviewer for the comments and suggestions.
Please note that in the replies below we are referring to the numbering of the lines, figures and references in the revised paper (docx file with Markup).
The research background and the method are described unclearly.
Sections “3. Methods” and “4. PCA-MRM model for TEC forecasting” are elaborated in the revised version. We hope now the math behind the model is clear.
There are so many papers performing the ionosphere EOF empirical models, which makes this paper less innovative. What’s more, the reviewer has soem main concerns as follows.
Yes, PCA is a widely used method to decompose variations of a studied parameter in the time-space or time-time (e.g., days vs hours) domains, which, however, doesn’t mean it cannot be used again. Same can be said about the linear regression. However, there are new ways to use old math. In fact, the approach we use to separate hourly and daily variations (with PCA) is rarely used in TEC modelling.
The TEC data are from one GNSS station’s observation. How to derive the TEC from raw GNSS observation? Is the average of vertical TECs from different satellite with certain cutoff elevation? How to calibrate the TEC with ROB TEC data? These important information are very limited in the manuscript, so it can be not determine the TEC data quality.
The dataset in question was published a couple years ago and a data paper was published as well [refs. 18-19]. The comparison between our TEC series and the ROB TEC was done in a paper published 2 years ago [ref. 20]. More details on the TEC data set and on the calibration procedure are added to ll. 127 and 140-148.
So many space weather parameters are considered in the regression process. But the possible influences on the ionospheric variation are not provided in the manuscript. At least the physical connections should be given in the manuscript.
The detailed analysis of the variations of TEC during most prominent storms of 2015 and due to the solar flares of 2015 was published 2 years ago [ref. 20].
Are the flare number with different levels necessary for the ionospheric forecasting?
This is an interesting question. In this paper we do not discuss in details the performance of different space weather parameters as regressors for the PCA-MRM model: we plan to publish a separate paper on this subject, however some preliminary results were presented at 17th European Space Weather Week 2021 (ESWW 2021) last year (“Space weather parameters as predictors in the TEC models, Anna Morozova, Teresa Barata, Tatiana Barlyaeva , at cd02 - Machine Learning and Statistical Inference Techniques”). The separate number of flares of different type were used exactly to understand if they are important for the model or not. We hope to be able to assess this issue in other paper.
Due to the different response time of the ionosphere to solar flux and geomagnetic activity, the fixed delay time interval is not suitable for the regression process.
The concept of the PCA-MRM model we propose allows only for the lags between space weather parameters (SWp) and TEC series that are multiple of a day, not hour, thus we cannot predict short term events such as response of the ionosphere to flares of the forecasted day. However, both our previous studies [ref. 20] and some works cited in Sec. 1 show that lags of 1-2 days works fine for the TEC forecasting purposes. Multiple flares lead to increase of the solar UV and XR fluxes that lasts for more than 1 day and can affect TEC daily amplitudes. The geomagnetic storms we studied have significant effect on the TEC variations during the day of a storm and the following days, more prolonged effects are rare, at least at middle latitudes we take into consideration. We tested a possibility to add a 3-days lag but the performance of the model deteriorated, thus we rejected lags longer than 2 days. Finally, the proposed lags and, what is more important, there combination, allow the forecasting skills of the PCA-MRM TEC models to be of the same order as other TEC models of comparable concept and inputs.
As the manuscript given: “All SWp series used in the PCA-MRM model have 1d time resolution”. I wonder what is the meaning of “mean lag1.2” , Is it the average TEC of the forecasting TEC with delay one day and with delay two day?
This is correct. As is mentioned in the text (ll. 280-282) “The 1h TEC series forecasts were made separately for the MRM models with lags of 1 and 2 days, and the final forecast is constructed as the arithmetic mean of these forecasts, hereafter PCA-MRM(L##, mean.lag1.2)”. The reason for this approach is now described in ll. 302-306

Round 2
Reviewer 1 Report
- Supplementary figures are still not readable, particularly the new ones, the font sixes are very small. Figures with closed boxes, proper legends, titles and good font size look good and enable the reader to understand the content more visually. One does not need to always read the entire caption to understand what the figure is about.
- The discussion on the rationale behind 1-2 days’ lag is still not sufficient. The reason, to the best of my understanding, is that a few SWp parameters are recorded at the end of the day and so on. Specifically, the explanation is not included in the revised version.
- If discussion of results of the current paper are discussed on “original paper”, what is the significance of this paper? Is this a companion paper? Overall, the current paper needs to stand by itself. Hence, the changes made to section 6 are required.
- Use of colloquial language in a scientific paper is not expected.
- Reply to comment on Table 2: Another careless mistake - …. 1 h and 1 d series….
- Lack of consistency is observed. MAE and |Delta TEC| are both used. Sometimes MAE is referred to as ‘Delta TEC’ (in supplementary figures). This is misleading.
- Comment on Line 485 (Version 1): “For all disturbed Days of 2015…..’. The answer is not satisfactory. No change is made in revised version. If the change is not made, it appears that the correlations are calculated for all days (quiet and disturbed)
- Figure S7 and S8: Still unreadable. However, the correction of “1.5” as “Lag1.2” is noted.
Author Response
Reply to Reviewer 1
We are very thankful to the Reviewer for the useful comments.
Please note that in the replies below we are referring to the numbering of the lines, figures and references in the revised paper (docx file with Markup, 2nd revision).
Supplementary figures are still not readable, particularly the new ones, the font sixes are very small. Figures with closed boxes, proper legends, titles and good font size look good and enable the reader to understand the content more visually. One does not need to always read the entire caption to understand what the figure is about.
Figure S7 and S8: Still unreadable. However, the correction of “1.5” as “Lag1.2” is noted.
author response: We made new SM files, now 4 pdfs with separate plots (please see the updated “Supplementary Materials” section), all plots fit the A4 size paper. New captions are made for Figs. S7-S8
The discussion on the rationale behind 1-2 days’ lag is still not sufficient. The reason, to the best of my understanding, is that a few SWp parameters are recorded at the end of the day and so on. Specifically, the explanation is not included in the revised version.
author response: As we already said in the Replies to the 1st Review, there are several reasons for using lagged SWp series:
- There are lagged correlations between SWp and TEC parameters
- The regression models with those lags perform well
- Such lags were used in other models
- Such lags allow us to predict TEC.
We added a corresponding sentence to Sec. 3.2.
If discussion of results of the current paper are discussed on “original paper”, what is the significance of this paper? Is this a companion paper? Overall, the current paper needs to stand by itself. Hence, the changes made to section 6 are required.
author response: By the “original paper” we meant “the original version of the paper, before the 1st revision”. As we already mentioned in the Replies to the 1st Review, some comparisons were already made in the text of the original version of the paper (in Sec. 6), however, following the comments of Reviewer 1, we added more discussion and comparison to other work to that section. All these changes were marked (red Markup) in the 1st revision.
Reply to comment on Table 2: Another careless mistake - …. 1 h and 1 d series….
author response: Captions of Tab. 2 are correct. If Reviewer means our text in the “Replies …”, we apologize for the typo.
Lack of consistency is observed. MAE and |Delta TEC| are both used. Sometimes MAE is referred to as ‘Delta TEC’ (in supplementary figures). This is misleading.
author response: There are 3 different parameters:
- Delta_TEC – difference between the observed and modelled series (l. 239) – 1h time resolution
- |Delta_TEC| – absolute value of the difference between the observed and modelled series (l. 239) – 1h time resolution
- MAE, the mean absolute error defined by eq. 4 as mean |Delta_TEC|. MAE can be calculated for a certain day or for all available days. The time intervals used to calculate MAE are mentioned in the text when needed.
Legends of Figs. S7d and S8b-c are corrected
Comment on Line 485 (Version 1): “For all disturbed Days of 2015…..’. The answer is not satisfactory. No change is made in revised version. If the change is not made, it appears that the correlations are calculated for all days (quiet and disturbed)
author response: The word “disturbed” is added to l. 540
